# A machine learning model for identifying patients at risk for wild-type transthyretin amyloid cardiomyopathy

Ahsan Huda[1], Adam Castaño[1], Anindita Niyogi[1], Jennifer Schumacher[1], Michelle Stewart[1], Marianna Bruno[1], Mo Hu[2], Faraz S. Ahmad[2], Rahul C. Deo [3] & Sanjiv J. Shah [2✉]

Transthyretin amyloid cardiomyopathy, an often unrecognized cause of heart failure, is now treatable with a transthyretin stabilizer. It is therefore important to identify at-risk patients who can undergo targeted testing for earlier diagnosis and treatment, prior to the development of irreversible heart failure. Here we show that a random forest machine learning model can identify potential wild-type transthyretin amyloid cardiomyopathy using medical claims data. We derive a machine learning model in 1071 cases and 1071 non-amyloid heart failure controls and validate the model in three nationally representative cohorts (9412 cases, 9412 matched controls), and a large, single-center electronic health record-based cohort (261 cases, 39393 controls). We show that the machine learning model performs well in identifying patients with cardiac amyloidosis in the derivation cohort and all four validation cohorts, thereby providing a systematic framework to increase the suspicion of transthyretin cardiac amyloidosis in patients with heart failure.

[1] Pfizer, Inc., New York, NY, USA. [2] Northwestern University Feinberg School of Medicine, Chicago, IL, USA. [3] Brigham and Women's Hospital, Boston, MA, USA. ✉email: sanjiv.shah@northwestern.edu

Cardiac amyloidosis, a prototypical infiltrative cardiomyopathy, is an increasingly recognized cause of heart failure (HF), particularly HF with preserved ejection fraction (HFpEF)[1,2]. Although once thought of as untreatable and associated with very poor outcomes, both of the main forms of cardiac amyloidosis (amyloidogenic light chain and transthyretin) are now treatable with chemotherapy and tafamidis (a transthyretin [TTR] stabilizer), respectively[1,3]. In addition, several imaging modalities —including speckle-tracking echocardiography[4], cardiac magnetic resonance[5], and bone scintigraphy[6]—are now available to assist with the non-invasive diagnosis of cardiac amyloidosis, including amyloidogenic TTR cardiomyopathy (ATTR-CM)[7]. Nevertheless, despite the availability of these treatments and diagnostic techniques, patients with cardiac amyloidosis continue to go undiagnosed or are diagnosed late in the course of the disease due to lack of awareness and recognition of amyloidogenic light chain and ATTR-CMs.

ATTR-CM may occur due to a variant in the *TTR* gene (hereditary ATTR-CM, most commonly the V122I variant, which is present in 3–4% of African Americans) or age-related misfolding of TTR (wild-type ATTR-CM). Patients with ATTR-CM, who develop symptomatic HF, have progressive decline in functional status and quality of life, and high morbidity and mortality with high rates of hospitalization and death[8]. Wild-type ATTR-CM appears to be more common than previously appreciated. In consecutive patients with HFpEF over the age of 60 years with a left ventricular wall thickness of 12 mm or greater screened with bone scintigraphy, the prevalence of wild-type ATTR-CM was 13%[9]. Furthermore, in consecutive patients undergoing transcatheter aortic valve replacement screened with bone scintigraphy, 16% had evidence of ATTR-CM[10]. ATTR-CM results from the dissociation of the normal tetrameric form of TTR into monomers that are prone to misfolding and aggregation, leading to TTR amyloid fibril formation and deposition in the myocardium[8].

Given the underdiagnosis and significant morbidity of ATTR-CM and availability of treatment with TTR stabilization, improved screening to enhance early diagnosis of the disease is essential, particularly for wild-type ATTR-CM for which no other means of systematic identification, such as genetic testing, is available. Clinical clues to the presence of ATTR-CM that have been described in the literature include bilateral carpal tunnel syndrome[11,12], lumbar spinal stenosis[11,13–15], and distal biceps tendon rupture[16]. However, additional predisposing or associated medical conditions may yet be discovered, which improve the likelihood of prompt diagnosis. We sought to develop and validate a machine learning model based on administrative medical claims data in the electronic health record (EHR), with the goal of creating a resource to facilitate the systematic screening and identification of patients with wild-type ATTR-CM. We subsequently validated the machine learning model in four additional cohorts. Here we show that the machine learning (ML) model performed well in identifying ATTRwt-CM vs. non-amyloid HF in the derivation cohort and all four validation cohorts, thereby providing a systematic framework to increase the suspicion of potential ATTRwt-CM in patients with HF.

## Results

This study comprised three parts as follows: (1) derivation (training) and testing of various supervised statistical learning models for the diagnosis of wild-type ATTR-CM in a large administrative medical claims dataset (IQVIA); (2) validation of the best-performing ATTR-CM model in additional large administrative medical claims and EHR datasets (IQVIA and Optum); and (3) testing of the final machine learning model in an EHR from a single, large healthcare system (Northwestern Medicine Enterprise Data Warehouse [NMEDW]) to observe the model's performance. The details of each of the five cohorts (the derivation cohort and the four validation cohorts) is listed in Supplementary Table 1.

The demographic and clinical characteristics of patients included in the five cohorts included in this study are summarized in Table 1. Patients in the first four cohorts were matched on age, sex, duration of medical history in the database, and number of healthcare visits, and patients in all five cohorts were only included if they were 50 years or older. The mean age of patients across the cohorts was 73–78 years old and the majority in the first four cohorts (>65%) were male based on the matching of non-amyloid HF controls to wild-type ATTR-CM or cardiac amyloidosis, the majority of whom were men, reflecting the known male predominance of ATTR-CM. In the NMEDW HF cohort (Cohort 5), which comprised all HF patients (not matched to ATTR-CM or cardiac amyloidosis), there were similar proportions of male and female patients. In all of the cohorts, comorbidities were common but differed between amyloid and non-amyloid HF patients. Patients with amyloid and HF were more likely to have diagnoses of atrial fibrillation and chronic kidney disease, and less likely to have hypertension, obesity, diabetes, and coronary artery disease. The total number of healthcare encounters and total duration of diagnostic history information in the datasets were similar across Cohorts 1–4 (IQVIA and Optum). The NMEDW cohort had the lowest value for both of these metrics.

**Model selection, performance, and validation.** Of the three statistical models tested (logistic regression, Random Forest, and XGBoost), the Random Forest model had the highest area under the receiver operating characteristic curve (AUROC) of 0.93 (vs. 0.91 for logistic regression and 0.90 for XGBoost), as shown in Supplementary Table 2. Thus, the Random Forest ML model was carried forward in all subsequent analyses. In the Cohort 1 test set, the model performed well in correctly predicting wild-type ATTR-CM HF vs. non-amyloid HF with an accuracy of 87% and AUROC of 0.93 (Table 2 and Fig. 1). We found that the optimal classification cut-point was an ATTR-CM probability threshold of 0.475 based on maximizing precision (positive predictive value [PPV]), recall (sensitivity), and accuracy metrics (Supplementary Fig. 1), which was used in subsequent analyses for Cohorts 2–4. The model was successfully validated in these three external nationally representative cohorts (Table 2 and Fig. 1a–c), although performance was better for predicting wild-type ATTR-CM compared to the more general cardiac amyloidosis diagnosis.

In Cohort 5, the NMEDW HF external validation dataset ($n = 261$ cardiac amyloidosis HF cases, $n = 39393$ non-amyloid HF controls), the ML model had an AUROC of 0.80 (Fig. 1d). The performance characteristics of the ML model in the NMEDW HF cohort varied based on the ML model-derived probability of the ATTR-CM diagnosis. Table 3 shows that as the probability cutoff for predicting ATTR-CM was increased from 0.5 to 0.75, sensitivity decreased from 64 to 11%, specificity increased from 85 to 99%, and accuracy increased from 84 to 99%. As shown in Table 4, the AUROC of the ML model in the NMEDW cohort performed better than models that include clinical factors such as age, sex, race, logBNP, and abnormal troponin levels. In addition, adding the ML algorithm-predicted probability of ATTR-CM to a base model that includes age, sex, race, and logBNP resulted in an integrated discrimination index of 0.039 (95% confidence interval [CI] 0.031–0.048), $P < 0.00001$ and a category-less net reclassification index of 0.25 (95% CI 0.22–0.28), $P < 0.00001$, both indicative that the addition of the ML algorithm improved

**Table 1 Demographic and clinical characteristics of the five cohorts included in the study.**

| Characteristic | Cohort 1: IQVIA (ATTR-CM) | | Cohort 2: Optum (ATTR-CM) | | Cohort 3: IQVIA (cardiac amyloid) | | Cohort 4: Optum (cardiac amyloid) | | Cohort 5: NMEDW* (cardiac amyloid) | |
|---|---|---|---|---|---|---|---|---|---|---|
| | ATTRwt-CM + HF (n = 1071) | Non-amyloid HF (n = 1071) | ATTRwt-CM + HF (n = 173) | Non-amyloid HF (n = 173) | Cardiac amyloid + HF (n = 7296) | Non-amyloid HF (n = 7296) | Cardiac amyloid + HF (n = 1943) | Non-amyloid HF (n = 1943) | Cardiac amyloid + HF (n = 261) | Non-amyloid HF (n = 39,393) |
| Age, years | 77 ± 7 | 77 ± 7 | 77 ± 8 | 74 ± 11 | 78 ± 8 | 77 ± 8 | 78 ± 9 | 77 ± 11 | 74 ± 9 | 72 ± 11 |
| Male | 84% | 84% | 82% | 81% | 67% | 67% | 68% | 69% | 69% | 56% |
| Ethnicity[a] | | | | | | | | | | |
| White | — | — | 66% | 80% | — | — | 72% | 80% | 54% | 71% |
| Black | — | — | 28% | 11% | — | — | 22% | 10% | 32% | 13% |
| Other | — | — | 6% | 9% | — | — | 6% | 10% | 14% | 16% |
| Location of final visit | | | | | | | | | | |
| Midwest | 28% | 20% | 47% | 48% | 24% | 21% | 49% | 49% | 100% | 100% |
| Northeast | 27% | 20% | 25% | 16% | 28% | 19% | 22% | 11% | 0% | 0% |
| South | 25% | 40% | 20% | 29% | 28% | 39% | 19% | 28% | 0% | 0% |
| West | 18% | 18% | 5% | 5% | 19% | 19% | 6% | 9% | 0% | 0% |
| Other/ unknown | 1% | 2% | 3% | 2% | 1% | 1% | 3% | 4% | 0% | 0% |
| Comorbidities | | | | | | | | | | |
| Hypertension | 90% | 96% | 83% | 88% | 89% | 92% | 81% | 86% | 72% | 78% |
| Obesity | 42% | 48% | 48% | 47% | 37% | 42% | 34% | 39% | 21% | 25% |
| Diabetes | 42% | 62% | 35% | 52% | 45% | 55% | 39% | 50% | 38% | 36% |
| CAD | 64% | 73% | 54% | 68% | 60% | 65% | 59% | 65% | 56% | 54% |
| CKD | 61% | 44% | 58% | 47% | 52% | 42% | 56% | 43% | 47% | 29% |
| Atrial fibrillation | 72% | 52% | 65% | 52% | 64% | 50% | 61% | 50% | 56% | 41% |
| Diagnosis history duration, years[b] | 9.6 (6.2-10.4) | 9.7 (6.3-10.5) | 6.2 (2.8-9.8) | 6.4 (3.0-9.5) | 8.7 (6.0-10.2) | 8.7 (6.0-10.2) | 5.2 (2.5-8.3) | 5.4 (2.7-8.2) | 6.2 (1.2-15.7) | 4.6 (0.9-10.1) |
| Number of visits[b] | 129 (82-210) | 131 (87-205) | 79 (34-187) | 76 (31-168) | 117 (63-193) | 112 (60-187) | 67 (25-141) | 66 (24-140) | 37 (9-99) | 21 (6-60) |

T-tests (for age), Wilcoxon rank-sum tests (for diagnosis history duration and number of visits), and χ²-tests (for categorical variables) were used to compare groups. P-values are two-sided. Adjustments were not made for multiple comparisons.
ATTRwt-CM wild-type amyloidogenic transthyretin cardiomyopathy, CAD coronary artery disease, CKD chronic kidney disease, NMEDW Northwestern Memorial Enterprise Data Warehouse.
[a]Ethnicity data were not available in the IQVIA data.
[b]Median (25th-75th percentile).
*P < 0.001 for age, sex, ethnicity, CKD, AF, diagnosis history, and number of visits, and P = 0.015 for hypertension for differences between cardiac amyloid and non-amyloid HF in the NMEDW cohort.

**Table 2 Validation of the machine learning model in four cohorts derived from medical claims and electronic health records.**

| Metric | Validation cohort | | | |
|---|---|---|---|---|
| | Cohort 1: IQVIA holdout (ATTR-CM) | Cohort 2: Optum (ATTR-CM) | Cohort 3: IQVIA (cardiac amyloid) | Cohort 4: Optum (cardiac amyloid) |
| Sensitivity, % | 87 | 90 | 56 | 61 |
| Specificity, % | 87 | 79 | 83 | 81 |
| PPV, % | 88 | 81 | 76 | 76 |
| NPV, % | 86 | 89 | 65 | 67 |
| Accuracy, % | 87 | 84 | 69 | 71 |
| ROC AUC | 0.93 | 0.95 | 0.76 | 0.78 |

*ATTR-CM amyloidogenic transthyretin cardiomyopathy, NPV negative predictive value, PPV positive predictive value, ROC AUC receiver operating characteristic area under the curve.*

performance of a model that used conventional clinical predictors alone. We also tested the performance of the Random Forest cardiac amyloidosis model in the NMEDW HF cohort and found that it performed better than the wild-type ATTR-CM model, with an AUROC of 0.81 (Supplementary Table 3). Within the NMEDW HF cohort, PPV was low, reflecting the low prevalence of cardiac amyloidosis in this cohort. However, as shown in Table 5, the high positive likelihood ratios associated with both the ATTR-CM and cardiac amyloidosis ML algorithms, combined with an estimated pre-test probability of 4% for ATTR-CM (based on a prior systematic screening study[9,20]), resulted in high post-test probabilities of ATTR-CM, which highlight its potential clinical utility in raising suspicion for the ATTR-CM diagnosis in HF patients.

**Clinical features associated with wild-type ATTR-CM.** The model output revealed several cardiac and non-cardiac clinical features that were associated with wild-type ATTR-CM. Supplementary Table 4 displays the prevalence and odds ratios for the top International Classification of Disease (ICD) code-based cardiac and non-cardiac phenotypes predictive of wild-type ATTR-CM. The relationship between prevalence and odds ratios for these phenotypes are displayed in Fig. 2. Besides HF and cardiomyopathy-related phenotypes, the strongest cardiac predictors included pericardial effusion/pericarditis, atrial flutter, cardiac conduction disorders, and abnormal serum enzymes, whereas the strongest non-cardiac predictors included carpal tunnel syndrome, synovitis/tenosynovitis, and ascites. Table 6 shows how combinations of diagnoses in claims data are associated with wild-type ATTR-CM. For example, one of the top phenotype combinations (by odds ratio) was the combination of joint disorders + osteoarthrosis + pleurisy or pleural effusion + HFpEF, which had a prevalence of 18.1% in wild-type ATTR-CM vs. 3.6% in non-amyloid HF (odds ratio 5.8 [95% CI 4.0–8.5], $p = 1.51 \times 10^{-28}$). Supplementary Table 5 shows expanded test characteristics for these phenotypic combinations. As shown in Fig. 3, the top non-cardiac phenotypes more commonly associated with wild-type ATTR-CM (vs. non-amyloid HF) preceded the cardiac phenotypes and the HF diagnosis by several years.

**Discussion**
Using medical claims data from a large, nationally representative database, we developed an ML prediction model for wild-type ATTR-CM, with subsequent validation and performance in multiple additional cohorts. Moreover, we identified antecedent

diagnoses and phenotypes associated with wild-type ATTR-CM, some of which have been previously reported in the literature and some not previously described, such as unique combinations of cardiac and non-cardiac phenotypes that were predictive of ATTR-CM. The results of our study are important, because our predictive model, if successfully employed within the EHR of healthcare systems, could lead to targeted testing and confirmation of ATTR-CM in at-risk patients, which could lead to earlier treatment of patients with wild-type ATTR-CM (and potentially other forms of cardiac amyloidosis) by raising the suspicion of the diagnosis in HF patients.

Cardiac amyloidosis is a specific cause of HF that is associated with high morbidity and mortality, and is often misdiagnosed as other, more common forms of HF and cardiac disease[21]. Disease-modifying treatments are now available for both of the most common types of cardiac amyloidosis (ATTR and amyloidogenic light chain), and earlier treatment in the course of disease may be associated with greater treatment response[22]. Therefore, earlier and more systematic diagnosis of cardiac amyloidosis is imperative. This is especially true for wild-type ATTR-CM, because currently there is no specific test (e.g., abnormal light chains or the presence of a genetic variant in the *TTR* gene), which is indicative of the potential disease in order to pursue a definitive diagnosis. Furthermore, identification of wild-type ATTR-CM is important, as we now recognize that the condition is more common than previously appreciated (e.g., 13% of patients with HFpEF over the age of 60 years with increased left ventricular wall thickness[9] and 16% of elderly patients undergoing transcatheter aortic valve replacement[10]).

ML models based on medical claims data for the prediction of diseases and phenotypes have been described in the medical literature with increasing frequency. Recent examples include ML models for opioid dependence, ankylosing spondylitis, drug-resistant epilepsy, prodromal Alzheimer's disease, and dementia[23–27]. ML studies vary in quality and often do not include the comprehensive validation and performance testing reported here[28,29]. Our study used both an internal test set and four external cohorts for validation.

In our study, model performance was slightly better in cohorts generated from EHR data (Optum) than in cohorts generated from medical claims data (IQVIA). This finding is likely explained by the fact that EHR data are more comprehensive than claims data. Machine learning models typically perform best using the data they learn from, but our model performed marginally better on Optum EHR data than in the IQVIA medical claims data (e.g., 91% vs. 87% accuracy in predicting wild-type ATTR-CM, although 71 and 71% accuracy in predicting cardiac amyloidosis). When applied in a single-center EHR setting (Cohort 5, NMEDW HF cohort), the model performed well overall with an AUROC of 0.80 and better model performance compared with demographic factors and biomarkers alone. Furthermore, when the Random Forest cardiac amyloidosis model was applied to this cohort, it performed even better with an AUROC of 0.85, which is not surprising, because the NMEDW HF cohort cases included all patients with a cardiac amyloidosis diagnostic code (not just ATTR-CM diagnoses). We excluded blood cancers and other non-ATTR-related amyloidosis codes from our cardiac amyloidosis cases to try to minimize the potential of overlap of our prediction model with other forms of cardiac amyloidosis (e.g., amyloidogenic light chain cardiac amyloidosis). Nevertheless, some patients with these other types of cardiac amyloidosis may have been included in the cases for Cohorts 3, 4, and 5. However, deriving and validating ML models for both the specific ATTR-CM diagnosis and the more general cardiac amyloidosis umbrella diagnostic code is important, because clinicians may use either term when diagnosing

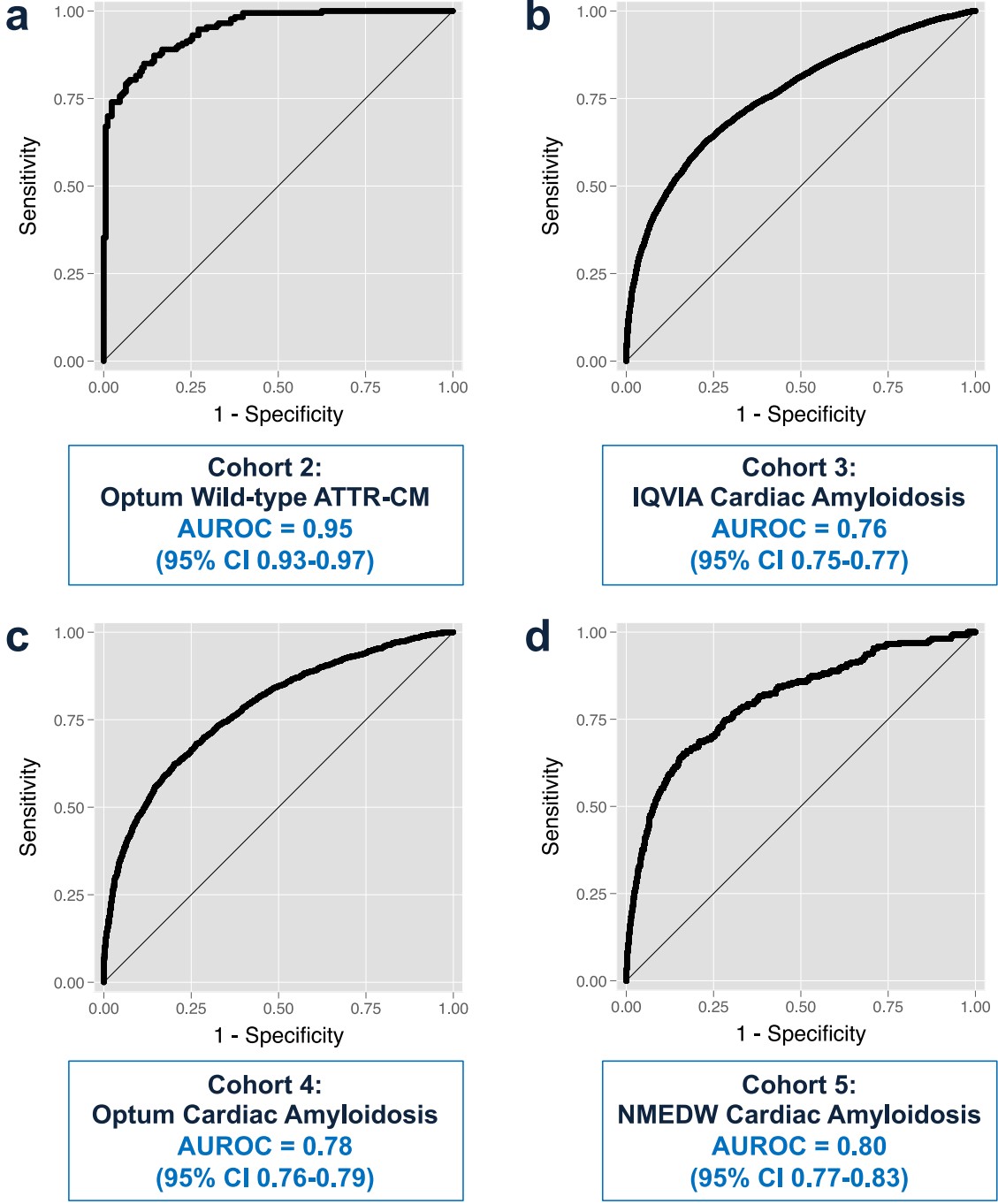

**Fig. 1 Receiver operating characteristic curves for the Random Forest machine learning model in the four validation cohorts. a** Optum ATTR-CM validation cohort. **b** IQIVA cardiac amyloidosis validation cohort. **c** Optum cardiac amyloidosis cohort. **d** Northwestern Medicine Enterprise Data Warehouse validation cohort. AUROC, area under the receiver operating characteristic curve; ATTR-CM, amyloidogenic transthyretin cardiomyopathy; NMEDW, Northwestern Medicine Enterprise Data Warehouse.

ATTR-CM. Given the potential overlap in the cardiac and extracardiac manifestations of the various types of cardiac amyloidosis, an ML model that predicts the presence of cardiac amyloidosis would still be clinically useful, because it would heighten clinical suspicion for the diagnosis, leading to further definitive testing, and a final diagnosis of whether cardiac amyloidosis is present or not. If present, the definitive testing would also determine the underlying type of cardiac amyloidosis.

As shown in Table 3, despite high overall accuracy, PPV was low because of the low prevalence of cardiac amyloidosis, which differed from the other development and validation datasets

(IQVIA and Optum), which were designed as case–control studies (with propensity score matching) and were not applied more broadly to an entire health system. In addition, the NMEDW HF cohort likely contains undiagnosed ATTR-CM patients, which can partially explain the low PPVs in this cohort, as several patients predicted as ATTR-CM are potentially undiagnosed and thus labeled as HF. Despite the relatively low PPVs, our ML model could still be useful clinically, because it would increase the suspicion of ATTR-CM, which could lead clinicians to re-evaluate routine tests performed in HF patients (e.g., electro-cardiography, echocardiography, and cardiac biomarkers), which

**Table 3 Prediction of cardiac amyloidosis in the Northwestern Medicine Enterprise Data Warehouse Heart Failure Cohort using the wild-type ATTR-CM Random Forest prediction model.**

| Metric | Probability cutoff for the diagnosis of ATTR-CM | | | | | |
|---|---|---|---|---|---|---|
| | >0.50 | >0.55 | >0.60 | >0.65 | >0.70 | >0.75 |
| Sensitivity, % | 69.7 | 64.0 | 52.5 | 36.8 | 22.2 | 11.1 |
| Specificity, % | 75.6 | 84.5 | 91.0 | 95.5 | 98.0 | 99.3 |
| PPV, % | 1.9 | 2.7 | 3.7 | 5.2 | 6.8 | 9.6 |
| NPV, % | 99.7 | 99.7 | 99.7 | 99.6 | 99.5 | 99.4 |
| Accuracy, % | 75.5 | 84.4 | 90.8 | 95.2 | 97.5 | 98.7 |
| LR+ | 2.86 | 4.12 | 5.85 | 8.24 | 11.07 | 15.97 |
| LR− | 0.40 | 0.43 | 0.52 | 0.66 | 0.79 | 0.90 |

ATTR-CM amyloidogenic transthyretin cardiomyopathy, LR+ positive likelihood ratio, LR− negative likelihood ratio, NPV negative predictive value, PPV positive predictive value.

**Table 4 Areas under the receiver operating characteristic curve for various prediction models in the Northwestern Medicine Enterprise Data Warehouse Heart Failure Cohort.**

| Model | N | AUROC |
|---|---|---|
| ATTRwt-CM RF model | 39,654 | 0.80 |
| ATTRwt-CM RF model, age > 70 years | 23,570 | 0.82 |
| Age only | 39,624 | 0.54 |
| Age + sex | 39,618 | 0.62 |
| Age + sex + ethnicity[a] | 39,203 | 0.70 |
| Age + sex + ethnicity + logBNP[b] | 20,419 | 0.73 |
| Age + sex + ethnicity + logBNP + abnormal troponin-I[c] | 15,046 | 0.73 |
| ATTRwt-CM RF model + age + sex + ethnicity | 39,203 | 0.83 |
| ATTRwt-CM RF model + age + sex + ethnicity + total number of encounters | 38,337 | 0.83 |

ATTRwt-CM amyloidogenic transthyretin (wild-type), AUROC area under the receiver operating characteristic curve, RF Random Forest, BNP B-type natriuretic peptide.
[a]Ethnicity categories: non-Hispanic White, non-Hispanic Black, Hispanic, Asian, others.
[b]Highest BNP value in the electronic health record, log-transformed.
[c]Based on the highest troponin-I in the electronic health record (abnormal defined as >0.04 ng/ml).

offer clues to the presence of ATTR-CM. Patients with corroborating findings on these routine clinical tests obtained in all HF patients (e.g., apical sparing on speckle-tracking echocardiography) could then undergo confirmatory non-invasive diagnostic testing (e.g., bone scintigraphy). Furthermore, even if a high probability threshold for the diagnosis of ATTR-CM was used (e.g., >0.75), as shown in Table 3, nearly 10% additional patients with ATTR-CM could be ultimately diagnosed, which would be an important clinical advance. As we have modeled previously, the pre-test probability of ATTR-CM is likely ~4% in HF patients[20]. Using this pre-test probability, Table 5 demonstrates the clinical utility of the ATTR-CM and cardiac amyloid ML models by showing how the high likelihood ratios result in large increases in post-test probability, thereby alerting clinicians to the possibility of the ATTR-CM diagnosis. Nonetheless, it is important to note that future prospective studies, with systematic application of gold standard tests for the diagnosis of ATTR-CM will need to be curated for further development and validation of our ML model.

Our findings on the cardiac and non-cardiac phenotypes predictive of wild-type ATTR-CM confirm what is known about the disease from the literature and clinical experience. For cardiac phenotypes, these include pericardial effusion/pericarditis, atrial arrhythmias, cardiac conduction disorders, and abnormal serum enzymes (likely indicative of abnormal elevations of troponin, which are common in cardiac amyloidosis); for non-cardiac phenotypes, these include carpal tunnel syndrome, bone and joint disorders, neuropathies, and soft tissue disorders. We also identified combinations of phenotypes that were highly associated with wild-type ATTR-CM and found that the non-cardiac diagnoses associated with ATTR-CM, as expected, were present for several years prior to the diagnosis of ATTR-CM, thus highlighting the opportunity for earlier diagnosis. In addition to corroborating earlier findings, our analysis reveals a comprehensive landscape of the manifestations ATTR disease that predate the diagnosis of ATTR-CM by ascribing the relative significance of disease characteristics (phenotypes) and combinations thereof.

ML represents an approach to automating increased identification of patients with diseases such as wild-type ATTR-CM. It is particularly useful, because it can efficiently evaluate complex interactions across multiple input predictors (e.g., ICD codes), a process that would be cumbersome using conventional statistical

**Table 5 Post-test probabilities for the Random Forest ATTR-CM and cardiac amyloid Random Forest models based on model performance in the Northwestern Medicine Enterprise Data Warehouse Heart Failure Cohort.**

| Model | Pre-test probability of ATTR-CM[a] | Random Forest model output cutoff for the diagnosis of ATTR-CM | LR+ | LR− | Post-test probability, LR+ | Post-test probability, LR− |
|---|---|---|---|---|---|---|
| Random Forest ATTR-CM model | 4% | >0.50 | 2.86 | 0.40 | 10.7% | 1.7% |
| | 4% | >0.55 | 4.12 | 0.43 | 14.8% | 1.8% |
| | 4% | >0.60 | 5.85 | 0.52 | 19.7% | 2.1% |
| | 4% | >0.65 | 8.24 | 0.66 | 25.7% | 2.7% |
| | 4% | >0.70 | 11.07 | 0.79 | 31.7% | 3.2% |
| | 4% | >0.75 | 15.97 | 0.90 | 40.1% | 3.6% |
| Random Forest cardiac amyloid model | 4% | >0.50 | 4.38 | 0.43 | 15.5% | 1.8% |
| | 4% | >0.55 | 7.13 | 0.53 | 23.0% | 2.2% |
| | 4% | >0.60 | 12.37 | 0.66 | 34.2% | 2.7% |
| | 4% | >0.65 | 21.78 | 0.79 | 47.8% | 3.2% |
| | 4% | >0.70 | 39.37 | 0.89 | 62.3% | 3.6% |
| | 4% | >0.75 | 72.18 | 0.96 | 75.2% | 3.9% |

The random forest ATTR-CM model was derived using diagnosis codes specifically for wild-type ATTR-CM. The random forest cardiac amyloid model was derived using the more nonspecific umbrella diagnosis code for cardiac amyloidosis.
ATTR-CM amyloidogenic transthyretin cardiomyopathy, LR+ positive likelihood ratio, LR− negative likelihood ratio.
[a]Pre-test probability was estimated to be 4% based on a prior publication (Kazi et al.[20]) that modeled the estimated prevalence of ATTR-CM in heart failure patients.

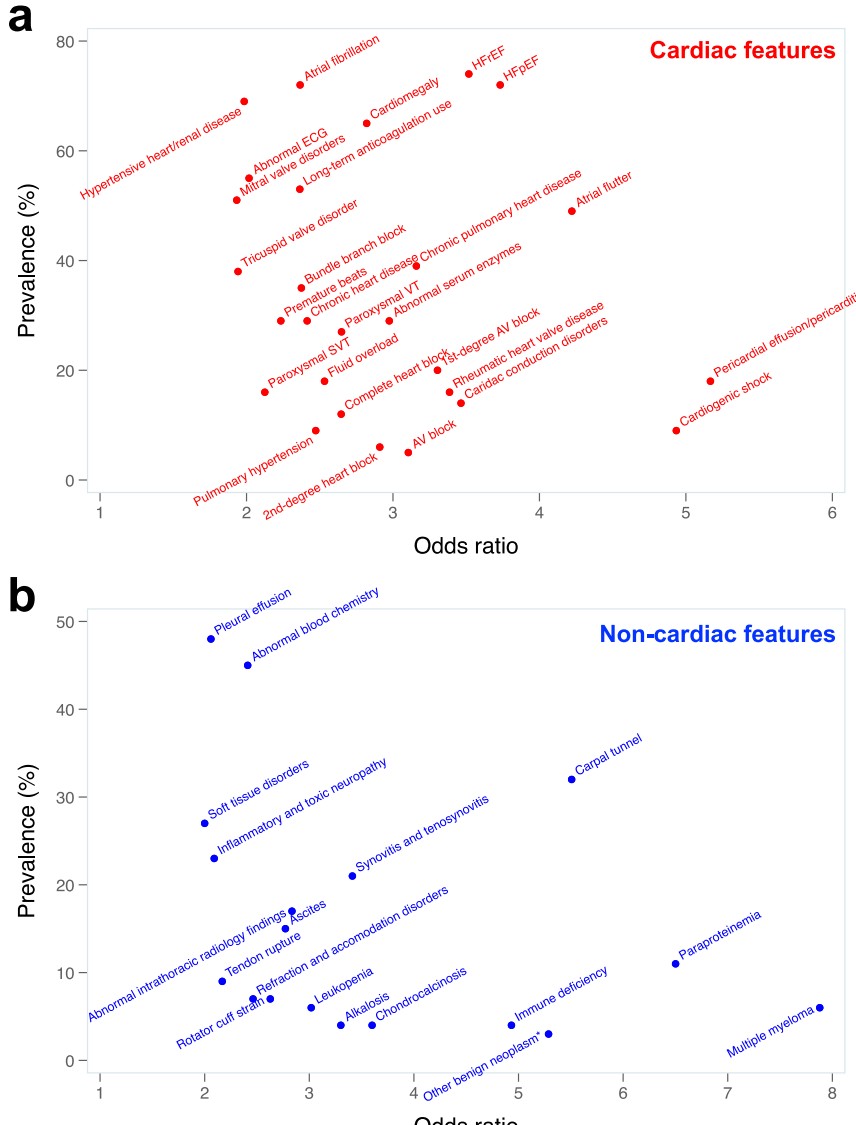

**Fig. 2 Odds ratio vs. prevalence for top clinical phenotypes predictive of wild-type ATTR cardiomyopathy. a** Cardiac phenotypes associated with wild-type ATTR cardiomyopathy. **b** Non-cardiac phenotypes associated with wild-type ATTR cardiomyopathy. All features associated with the diagnosis of ATTR cardiomyopathy at a significance level of $P < 10^{-4}$, which had an odds ratio (OR) < 10, were included in the graphs. The three features that had an OR > 10 that met the *p*-value threshold were: hypertrophic cardiomyopathy (OR 15.8, prevalence 11%); localized adiposity (OR 26.6, prevalence 2%); and organ transplantation (OR 23.4, prevalence 4%). Some diagnoses that were associated with ATTR cardiomyopathy (e.g., hypertrophic cardiomyopathy, multiple myeloma) were likely initial misdiagnoses, as these diagnoses (similar to all diagnoses included here) preceded the ATTR cardiomyopathy diagnosis. Univariate logistic regression was used to calculate odds ratios. *Localized to the connective tissue or soft tissue. AV, atrioventricular; ECG, electrocardiogram; HFrE, heart failure with reduced ejection fraction; HFpEF, heart failure with preserved ejection fraction; SVT, supraventricular tachycardia; VT, ventricular tachycardia.

approaches. Nonetheless, as we have shown here, it is important to demonstrate that ML approaches outperform conventional statistical modeling, and that a variety of ML models are tested to determine the best-performing model. In addition to finding undiagnosed patients, ML models may help find patients earlier in their diagnostic journey and can reveal a broader picture of the signs and symptoms of the disease, allowing a deeper understanding of its pathogenesis and pathophysiology.

ML models, such as the one that we developed, can be used to calculate the probability of wild-type ATTR-CM in HF patients and elevate the index of suspicion, after which further confirmatory testing can be done, thereby enhancing diagnosis and clinical recognition of wild-type ATTR-CM, which is increasingly important given the high morbidity and mortality of wild-type

ATTR-CM and the availability of tafamidis, a disease-modifying therapy. Importantly, such models can pull requisite data from the EHR and can be deployed in a fully automated manner, thus performing the initial step in disease detection on a broad scale. Further machine learning-based automated analyses of electrocardiogram tracings[30] and echocardiographic images[31] (which are standard clinical tests in HF patients) could also be used to heighten diagnostic suspicion of cardiac amyloidosis and ATTR-CM. Nevertheless, the potential for improved early diagnosis of wild-type ATTR-CM with incorporation of these ML algorithms must be balanced with the potential cost implications and downstream effects of false positives.

The work reported here has clinical applicability not only for wild-type ATTR-CM but also other diseases that are

**Table 6 Top 21 combinations of phenotypes based on ICD codes and their association with wild-type ATTR cardiomyopathy (IQVIA dataset, Cohort 1).**

| Combination of phenotypes[a] | Prevalence in ATTR-CM | Prevalence in controls | Number of phenotypes in addition to HF | Odds ratio (95% CI) | P-value |
|---|---|---|---|---|---|
| Combined systolic and diastolic HF, HFpEF | 52.1% | 17.3% | 2 | 5.2 (4.2–6.3) | $2.02 \times 10^{-65}$ |
| Carpal tunnel syndrome | 31.9% | 7.8% | 1 | 5.5 (4.2–7.1) | $4.62 \times 10^{-46}$ |
| AF, joint disorders, HFpEF | 29.7% | 7.0% | 3 | 5.6 (4.2–7.4) | $1.11 \times 10^{-43}$ |
| Heart block, cardiomegaly, HFpEF | 28.7% | 6.4% | 3 | 5.8 (4.4–7.7) | $1.54 \times 10^{-43}$ |
| Cardiomegaly, joint disorders, HFpEF | 28.7% | 6.5% | 3 | 5.7 (4.3–7.6) | $4.70 \times 10^{-43}$ |
| Heart block, CKD, HFpEF | 26.6% | 6.6% | 3 | 5.1 (3.8–6.8) | $9.97 \times 10^{-37}$ |
| AF, cardiomegaly, soft tissue disorders, HFpEF | 24.1% | 5.7% | 4 | 5.2 (3.9–7.1) | $2.95 \times 10^{-34}$ |
| Heart block, soft tissue disorders, HFpEF | 23.7% | 5.7% | 3 | 5.1 (3.8–7.0) | $3.14 \times 10^{-33}$ |
| AF, cardiomegaly, joint disorders, combined systolic and diastolic HF | 23.2% | 5.2% | 4 | 5.4 (4.0–7.5) | $2.99 \times 10^{-34}$ |
| Heart block, cardiomegaly, joint disorders | 22.0% | 5.1% | 3 | 5.2 (3.8–7.2) | $2.28 \times 10^{-31}$ |
| Heart block, joint disorders, combined systolic and diastolic HF | 21.8% | 4.7% | 3 | 5.6 (4.1–7.9) | $5.88 \times 10^{-33}$ |
| Cardiomegaly, joint disorders, soft tissue disorders, combined systolic and diastolic HF | 21.5% | 5.0% | 4 | 5.1 (3.7–7.1) | $2.63 \times 10^{-30}$ |
| Joint disorders, osteoarthrosis, pleurisy or pleural effusion, combined systolic and diastolic HF | 18.6% | 4.2% | 4 | 5.2 (3.7–7.4) | $6.63 \times 10^{-27}$ |
| AF, joint disorders, pleurisy or pleural effusion, combined systolic and diastolic HF | 18.3% | 3.9% | 4 | 5.4 (3.8–7.9) | $1.42 \times 10^{-27}$ |
| Joint disorders, osteoarthrosis, pleurisy or pleural effusion, HFpEF | 18.1% | 3.6% | 4 | 5.8 (4.0–8.5) | $1.51 \times 10^{-28}$ |
| AF, CKD, pleurisy or pleural effusion, soft tissue disorders, combined systolic and diastolic HF | 16.0% | 3.5% | 5 | 5.1 (3.5–7.6) | $2.96 \times 10^{-23}$ |
| AF, heart block, CKD, pleurisy or pleural effusion, combined systolic and diastolic HF | 15.3% | 3.3% | 5 | 5.3 (3.6–8.0) | $5.96 \times 10^{-23}$ |
| AF, heart block, CKD, soft tissue disorders, combined systolic and diastolic HF | 15.1% | 3.4% | 5 | 5.1 (3.5–7.6) | $5.44 \times 10^{-22}$ |
| AF, heart block, joint disorders, osteoarthrosis, soft tissue disorders | 14.5% | 3.2% | 5 | 5.1 (3.5–7.7) | $3.19 \times 10^{-21}$ |
| AF, CKD, pleurisy or pleural effusion, soft tissue disorders, HFpEF | 14.4% | 3.2% | 5 | 5.1 (3.4–7.7) | $5.57 \times 10^{-21}$ |
| Heart block, CKD, pleurisy or pleural effusion, soft tissue disorders, combined systolic and diastolic HF | 11.8% | 2.5% | 5 | 5.1 (3.3–8.1) | $1.62 \times 10^{-17}$ |

Univariate logistic regression was used to calculate odds ratios. P-values are two-sided. Adjustments were not made for multiple comparisons. AF atrial fibrillation (includes atrial flutter), CKD chronic kidney disease, HF heart failure, HFpEF heart failure with preserved ejection fraction.
[a]One or more of these combinations were present in 876 (82%) of the IQVIA Cohort 1 patients with wild-type ATTR-CM.

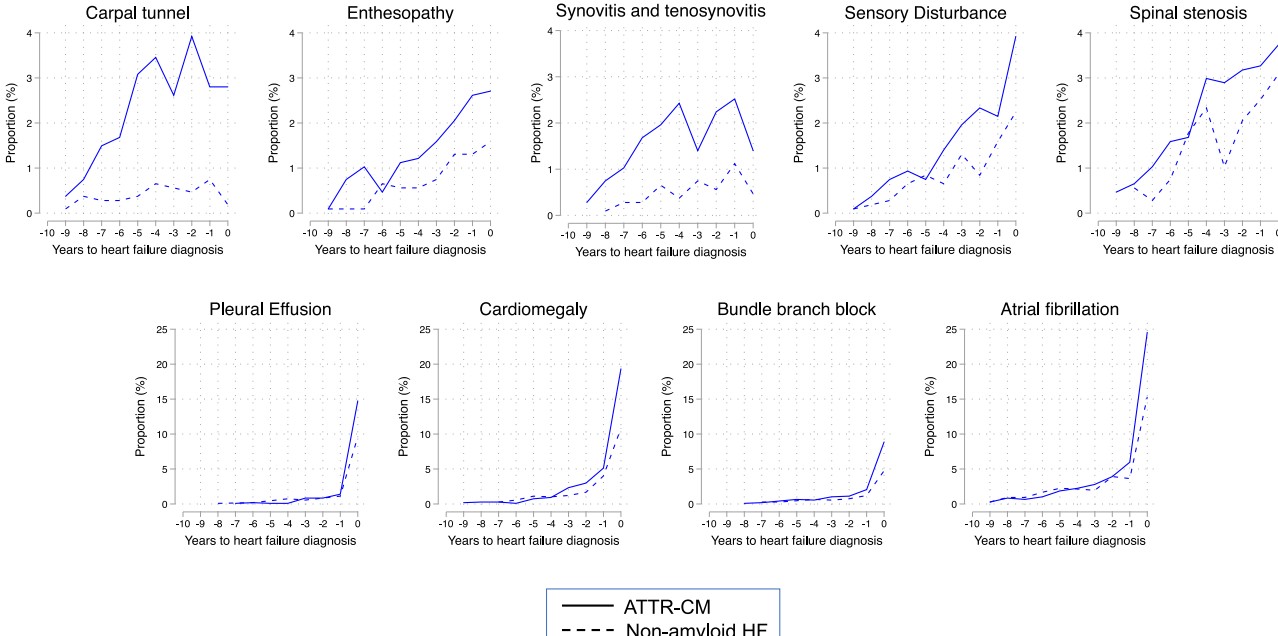

**Fig. 3 Time course of non-cardiac and cardiac phenotypes associated with wild-type ATTR cardiomyopathy vs. non-amyloid heart failure prior to the diagnosis of heart failure.** The proportion of patients at each time point (years before heart failure diagnosis) with a first diagnosis of an associated feature (phenotype). The cumulative proportion of patients with each particular phenotype is equal to the sum of the proportions from each of the years preceding the heart failure diagnosis. ATTR-CM, amyloidogenic transthyretin cardiomyopathy.

underdiagnosed or misdiagnosed. The analytical approach we used has broad applicability, because ICD is the international language of medical diagnoses and its ease of use in the EHR setting. Possible clinical applications range from clinical decision support via deployment at hospital EHRs as alerts for physicians or web-based risk calculators. It can also provide a systematic framework for understanding various constellations of signs and symptoms, especially for rare or under-recognized diseases.

Strengths of our study include the development of a prediction model for wild-type ATTR-CM in a large medical claims database, with validation and testing in multiple additional cohorts. We also tested the performance of our model in an independent external cohort within a single healthcare system. In addition, we performed comprehensive mapping of ICD codes to phenotypes, which allowed us to investigate the associated signs and symptoms that may predict and predate the diagnosis of wild-type ATTR-CM. The ability to convert the predictive signals (i.e., ICD codes [features]) from our model output into more readily understandable phenotypes with odds ratios for their association with wild-type ATTR-CM is an additional strength of our analytical approach. Although many of the identified phenotypes/features associated with wild-type ATTR-CM that we identified in the medical claims data are already well described in the literature, a strength of our ML approach is the ability to automatically identify patterns suggestive of ATTR-CM, which may not be obvious to the clinician, because they are derived using data from multiple diagnostic codes across several different disease domains and organ systems.

Several limitations should be considered when interpreting the results of our study. First, our study is limited by the use of ICD codes to identify HF patients and as the gold standard for assigning cases and controls. HF, a broadly defined clinical syndrome, may be inaccurately coded in the medical record. In addition, because of the nature of the datasets used for our analysis and the fact that diagnosis codes do not capture electrocardiographic voltage or echocardiographic markers, we do not have data on these phenotypes or additional laboratory data

(such as TTR concentrations [measured clinically as pre-albumin]), which could have assisted in the assignment of cases and controls. Therefore, in some of the non-amyloid HF controls, it is possible that cardiac amyloidosis is present but simply undiagnosed. However, given the low prevalence of wild-type ATTR-CM, gold-standard evaluation of cases and controls would only be feasible in highly selected cohorts and tertiary referral hospitals, which would reduce the generalizability of any derived predictive model. Second, our model was constructed on the basis of an ICD code specific to wild-type ATTR-CM. This diagnostic code is relatively new and may not be used universally; thus, biases could have been introduced by individual, institutional, or regional ICD coding behaviors that limit the applicability of our model. For this reason, we also tested the ML model on cases defined by the more general cardiac amyloidosis term and we also developed a specific ML model for the cardiac amyloidosis term. Furthermore, control patients with HF without the diagnosis could still have undiagnosed wild-type ATTR-CM. Nonetheless, we were able to validate our model in cohorts that included the more general ICD codes of cardiac amyloidosis and any misdiagnosis in our non-amyloid HF controls would have reduced our model's performance. Any curated dataset with ATTR-CM and controls will introduce bias into the performance of our ML model; thus, we chose to apply our model to the setting in which it would be used clinically (the independent EHR testing cohort [NMEDW]). In this cohort, we did not prospectively perform tissue biopsy or bone scintigraphy for wild-type ATTR-CM and blood testing for amyloidogenic light chain amyloidosis, which would have validated these diagnoses. This type of prospective study could be done in the future as our model is applied in clinical practice. Third, AUROC may be misleading for relatively rare (i.e., low prevalence) diagnoses such as wild-type ATTR-CM. However, we report a wide range of model performance indices in addition to the AUROC and we include positive and negative likelihood ratios (Table 5), which are useful for the clinical interpretation of our results. Finally, our model should not be viewed as definitive for the diagnosis of either ATTR-CM or

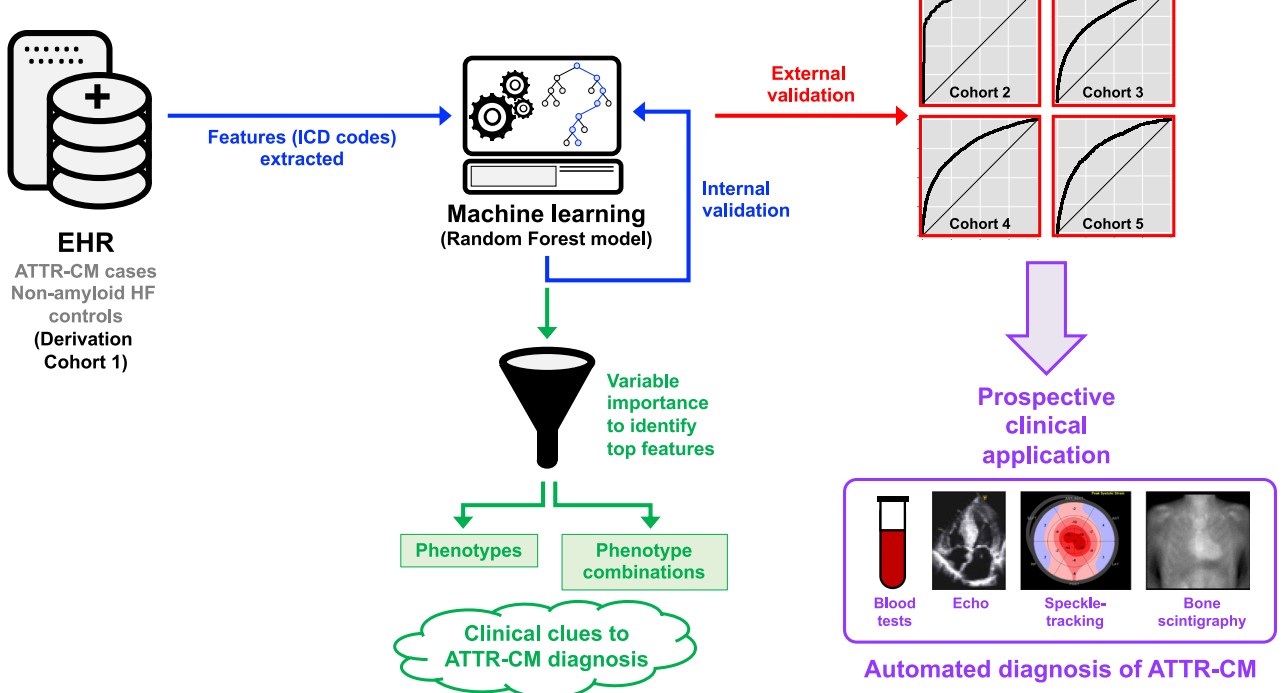

**Fig. 4 Development and validation of a machine learning model of medical claims data for the systematic identification of wild-type transthyretin amyloid cardiomyopathy.** Nationally representative medical claims data were used to develop a cohort of ATTR-CM and non-amyloid HF controls. ICD codes were extracted and used as features to train a Random Forest machine learning model, which was then internally tested in the derivation cohort. The model was then validated in four external cohorts, one of which was a single health system that is similar to how the model would be used in the clinical setting. The top features (ICD codes) based on variable importance in the Random Forest model were also used to generate phenotypes and phenotype combinations associated with the ATTR-CM diagnosis, which provide clinical insight and clues into the diagnosis. In the future, additional prospective clinical validation with blood tests, echocardiography (with speckle-tracking strain analysis), and bone scintigraphy can be used to verify the ATTR-CM diagnosis with the ultimate goal to automate the identification of ATTR-CM, thereby leading to earlier diagnosis and intervention of these high-risk patients. ATTR-CM, amyloidogenic transthyretin cardiomyopathy.

cardiac amyloidosis, but instead could be used as a starting point for identifying at-risk individuals for further evaluation with speckle-tracking echocardiography, cardiac magnetic resonance imaging with contrast, bone scintigraphy, and/or blood tests for amyloidogenic light chain amyloidosis. Subsequent evolution and simplification of this ML algorithm might further be clinically applicable at the individual patient level, for instance, in a HF clinic setting, to help assess probability and likelihood of ATTR-CM among patients presenting with undifferentiated HFpEF.

We have developed and validated a machine learning approach based on medical claims data, which provides a systematic framework for the screening and identification of potential wild-type ATTR-CM patients (Fig. 4). Despite the constraints of using medical claims data to build our prediction model, we illustrated that the machine learning model may be potentially useful in identifying wild-type ATTR-CM patients and could work in the background in EHR systems to automatically calculate probability scores and flag HF patients that may benefit from further evaluation for ATTR-CM. We also confirmed the clinical profile of wild-type ATTR-CM patients consistent with published literature and provide additional insights to aid in identifying undiagnosed patients. Application of models leveraging large datasets with ML can serve as an example for other rare or underdiagnosed diseases.

## Methods

**Dataset curation.** The primary dataset used for training and testing the model was sourced from IQVIA, Inc. (Durham, NC), and was composed of medical claims for patients with ICD diagnosis codes for HF and amyloidosis (listed in Supplementary Table 5). The use of a date range (observation window) of 1 January 2008 to 30 April 2019 yielded a dataset of ~300 million US patients. If an ICD-9 and/or ICD-

10 code was present in the claim, it was extracted and added to a list of diagnoses for each patient. A secondary dataset used for validating the model was sourced from the Optum® de-identified Electronic Health Record dataset (Optum, Inc., Eden Prairie, MN) and comprises clinical information derived from EHRs. The date range (observation window) of 1 January 2008 to 31 December 2018 resulted in a dataset of ~88 million US patients. Patient diagnosis history in the form of ICD-9 and/or ICD-10 codes was extracted from the diagnosis table within the EHR dataset and was added to a list of diagnoses for each patient. Use of the IQVIA and Optum datasets was approved by the Northwestern University Institutional Review Board (IRB). Informed consent was waived under Northwestern University's IRB regulations, as these are de-identified datasets. The use of IQVIA and Optum patient data (model training and initial validation portions of the study) was performed in accordance with the Declaration of Helsinki.

Prior to training the machine learning algorithms, all ICD-9 codes were converted to ICD-10 codes. Only the presence or absence of the ICD codes (and not the temporal sequence of ICD codes) were included as features for training the machine learning algorithm.

**Cohort generation.** Supplementary Table 5 lists all of the cohorts used for model training, testing, and validation, and includes a complete set of rules that were used to create these cohorts. Two sets of case/control cohorts of patients were created for model training from the IQVIA data. Wild-type ATTR-CM cases included patients with a diagnosis code specific to this type of amyloidosis (ICD-10 code E85.82) along with a diagnosis code for HF; controls included patients diagnosed with HF, who did not have a diagnosis of any type of amyloidosis or amyloid-related conditions. For each patient in the wild-type ATTR-CM cohort, a patient was selected from the non-amyloid HF cohort using a propensity score matching algorithm, with age, gender, duration of medical history in the database, and number of healthcare visits as matching parameters. An additional case/control cohort for validating the model was created for patients with cardiac amyloidosis; patients with organ-specific amyloidosis (ICD-10 code E85.4) and HF were selected and 1 : 1 propensity-matched with non-amyloid HF controls. Propensity score matching was performed in R using the MatchIt package.

Cardiac amyloidosis is an umbrella term that includes the diagnosis of wild-type ATTR-CM, hereditary ATTR-CM, and amyloidogenic light chain amyloidosis with

cardiac involvement. Whether a patient with ATTR-CM is coded as wild-type ATTR-CM or the more general term of cardiac amyloidosis can vary based on clinical practice patterns and billing practices. For this reason, we evaluated model performance in both wild-type ATTR-CM and cardiac amyloidosis validation cohorts (see the Supplementary Materials for further details). Within the cohorts that used the general cardiac amyloidosis term (Cohorts 3, 4, and 5), when defining cases we excluded blood cancers (reflective of amyloidogenic light chain cases) and other non-ATTR-CM-related amyloidosis terms, with the goal of capturing cases that had wild-type or hereditary ATTR-CM but were coded as the more general cardiac amyloidosis term.

**Feature selection**. We identified all ICD-9 and ICD-10 codes that were present in at least 2% of wild-type ATTR-CM patients in the IQVIA medical claims data. Each of these codes were then used as potential features for the machine learning model. Features derived from ICD codes of any amyloidosis condition (i.e., ICD-9 277.30 or ICD-10 E85) and a specific code for "cardiomyopathy in diseases classified elsewhere" that tends to accompany a cardiac amyloid diagnoses were removed, to ensure no information relevant to the diagnosis of ATTR-CM or cardiac amyloidosis was "leaked" to the machine learning model. Model features were created by using a hierarchical mapping of ICD codes at the Sub Chapter (diagnosis category), Major (diagnosis name), and Short Description (diagnosis description) levels, obtained from the ICD Data R package and derived from the 2016 release of ICD-10-CM. New codes added after 2016 used the 2019 release of ICD-10-CM.

**Model training and testing**. All machine learning analyses were performed in Python (using the numpy, pandas, sklearn, matplotlib, and GridSearchCV packages). We tested three supervised algorithms (logistic regression, XGBoost, and Random Forest) on the training input dataset with a train-to-test split ratio of 80:20 using the aforementioned dataset (Cohort 1), with each patient labeled as wild-type ATTR-CM ICD-10 code (E85.82) present or absent (non-amyloid HF control). Within the training set, fivefold cross-validation was used to select hyperparameters. A grid search algorithm was implemented to determine the best parameters for each of the three algorithms. Grid search uses heuristic methods to estimate the optimal set of hyperparameters for machine learning algorithms[17]. The types and ranges of hyperparameters used for grid search were as follows: number of trees: 200, 300, 400, 500; maximum depth of tree: 5, 10, 15, 20, none; minimum samples per leaf: 2, 3, 4, 5; minimum samples split: 2, 3, 5, 7; bootstrap: true, false; and cross-validation (K fold): 3-fold.

Testing was done in the 20% holdout sample and the following parameters were calculated based on the number of true positives, true negatives, false positives, and false negatives: sensitivity, specificity, positive and negative predictive values, and accuracy. The Random Forest model, which had the best diagnostic performance (as evaluated by the AUROC curve), was carried forward for further validation, as described below. In separate analyses, we repeated the aforementioned process to derive a model of the more general diagnosis of cardiac amyloidosis as opposed to wild-type ATTR-CM (see the Supplementary Materials for further details).

**Model validation**. The model derived from the IQVIA wild-type ATTR-CM population (Cohort 1) was validated in three other cohorts (Cohorts 2, 3, and 4, described in Supplementary Table 5) to evaluate its generalizability to a broader group of populations. We again computed sensitivity, specificity, positive and negative predictive values, and accuracy.

To further examine clinical generalizability of the ML model, we performed additional external testing of the derived and validated wild-type ATTR-CM Random Forest machine learning model in the NMEDW EHR cohort[18]. Use of the NMEDW dataset was approved by the Northwestern University IRB. Waiver of informed consent was granted under Northwestern University's IRB regulations, as this was a chart review analysis. The use of NMEDW patient data (model validation portion of the study) was performed in accordance with the Declaration of Helsinki. The NMEDW dataset was queried for all patients with any ICD-9 or ICD-10 HF diagnosis code using a date range of 1 June 2009 to 31 May 2019. Only patients aged ≥50 years were included. Cases were patients with an organ-limited amyloidosis code (E85.4) plus a HF diagnosis code but not blood cancer, light chain amyloidosis, end-stage renal disease, cerebral amyloid angiopathy, or intracranial hemorrhage diagnoses. Non-amyloid HF controls were patients with a HF code but none of the amyloidosis codes. These were the same definitions used in Cohorts 3 and 4, with the exception that there was no matching of cases and controls (all non-amyloid HF patients were included in the analysis to mimic clinical application of the ML algorithm). Supplementary Fig. 1 displays a flowchart detailing the inclusion/exclusion criteria for cases and controls. The Random Forest model was then applied to the NMEDW cohort and the probability of wild-type ATTR-CM was calculated for each patient. For all patients in the NMEDW HF cohort, we constructed 2 × 2 tables to determine true positives, true negatives, false positives, and false negatives, to calculate sensitivity, specificity, positive and negative predictive values, and accuracy. We also generated ROC curves for the performance of the Random Forest model in the NMEDW HF cohort (age ≥50 years) and we computed AUROC for these curves and compared them to additional models (age, age + sex, age + sex + race, age + sex + race + logBNP, age + sex + race + logBNP + abnormal troponin-I (>0.04 ng/ml), age + sex + race

+ Random Forest model, and age + sex + race + Random Forest model + total number of encounters).

**Mapping ICD codes to phenotypes**. Once we identified features (ICD codes) that were predictive of the wild-type ATTR-CM diagnosis, we sought to better understand the clinical meaning of these features. Therefore, we used a phenotype grouping system by combining one or more related ICD codes into distinct diseases or traits (based on PheWAS Phecode v.1.2)[19]. ICD codes mapped to multiple phenotypes were de-duplicated via selection of the single most clinically relevant phenotype.

**Associations of phenotypes with wild-type ATTR cardiomyopathy**. Of the features (ICD codes) included in the Random Forest model, those that were found to have some level of variable importance for the diagnosis of wild-type ATTR-CM (>97%) were mapped to phenotypes based on PheWAS mapping described above. Next, we tested the associations of each of these phenotypes with the wild-type ATTR-CM diagnosis in Cohort 1. We used logistic regression analyses to compare individual phenotypes between wild-type ATTR-CM cases and non-amyloid HF controls, and odds ratios (with 95% confidence intervals) were calculated. Correction for multiple tests was performed using the Bonferroni method. A two-sided $p$-value < 0.05 after Bonferroni correction was considered statistically significant.

Next we examined combinations of phenotypes that are associated with wild-type ATTR-CM. The Random Forest model derives its predictive power from specific interactions (trees) of features associated with wild-type ATTR-CM. We sought to elucidate these trees by identifying combinations of phenotypes with high predictive value. This was done by selecting ten phenotypes with the highest odds ratios and deriving all combinations of up to five phenotypes. We used the Python Scikit-learn set cover algorithm to identify the top phenotypes or phenotype combinations with odds ratios > 5 for further interrogation for their ability to identify the maximum number of ATTR-CM patients within the IQVIA wild-type ATTR-CM cohort (Cohort 1).

Finally, we mapped the time course for the most common phenotypes in relation to the first documented diagnosis of wild-type ATTR-CM, to help delineate the temporal aspect of features in relation to the documented diagnosis of ATTR-CM.

**Reporting summary**. Further information on research design is available in the Nature Research Reporting Summary linked to this article.

## Data availability

The IQVIA training and validation datasets have been made available for download (see Source Data). The Optum and Northwestern Medicine Enterprise Data Warehouse datasets used for this study could not be made available publicly due to data use agreements and the possibility for identification of individual patients, respectively, but will be made available to qualified investigators upon request with evidence of institutional review board approval. Source data are provided with this paper.

## Code availability

The code used for training and validation of the models (including instructions for use of the code and notations for the software needed to run the code) have been made available for download (see Source Data).

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

## Acknowledgements

This study was supported by Pfizer. This work was also supported by grants from the National Institutes of Health (R01 HL140731, R01 HL120728, R01 HL107577, and R01 HL149423); the American Heart Association (#16SFRN28780016, #15CVGPSD27260148, One Brave Idea, Apple Heart and Movement Study); the Agency for Healthcare Research and Quality (K12 HS026385); and the Patient-Centered Outcomes Research Institute and a contract from the People-Centered Research Foundation (#1236). Tracking of co-author contributions and data checks were provided by Donna McGuire and Joshua Fink, PhD, of Engage Scientific Solutions and funded by Pfizer; no contribution was made to editorial content.

## Author contributions

A.H., A.C., J.S., M.S., M.B., R.C.D., and S.J.S. conceived of and designed the work. A.H. and F.S.A. acquired the data. A.H., A.N., M.H., and S.J.S. performed statistical and computational analyses. S.J.S. drafted the work. All authors participated in interpretation of the data and substantively revised the work.

## Competing interests

S.J.S. has received grants from Actelion, AstraZeneca, Corvia, Novartis, and Pfizer; and has received consulting fees from Actelion, Amgen, AstraZeneca, Bayer, Boehringer-Ingelheim, Cardiora, Eisai, Ionis, Ironwood, Merck, Novartis, Pfizer, Sanofi, and United Therapeutics. R.C.D. has received a grant from GE Healthcare and has received consulting fees from Novartis and Pfizer. A.C., M.B., A.H., A.N., M.S., and J.S. are full-time employees of Pfizer. All other authors report no competing interests.
