## [Peer Review File · Nature Communications]

REVIEWER COMMENTS

Reviewer #2 (Remarks to the Author):

Development and Validation of a Machine Learning Model of Medical Claims Data for the Systematic Identification of Wild-Type Transthyretin Amyloid Cardiomyopathy

The study population, feature selection, and machine learning methodology used in this paper are very robust. The reviewers' comments have been somewhat addressed. It would be helpful to see changes made in the paper, this is currently missing. There are still major concerns which should be addressed throughout the manuscript carefully.

Major concerns:

1) The clinical relevance of all this is unclear. The argument is that amyloid is often missed or not considered as an etiology of heart failure, and that there is no specific test for amyloid which is true. However wild-type amyloid is more often an indolent disease with a slower course, and it's not clear why this was specifically chosen to examine from the vast database.

2) Also, absence of monoclonal protein in serum combined with a positive bone scintigraphy can now diagnose wild type ATTR with 100% specificity and PPV. The authors also fail to capitalize on their findings in their final discussion paragraph, as ultimately this machine learning model could work in the background to calculate prediction scores and flag patients with heart failure on EHR systems, and there is only brief mention of this.

3) ICD coding for wild-type amyloid was used as the gold standard – this is flawed as we do not know what the specific ECG, imaging, and histology findings are for these patients, or what criteria was used to make this diagnosis. Even within wild-type amyloid there are many different genetic variants all with different phenotypes, and the authors fail to tease this out as their ICD coding was purely "wild-type". This study seems adequately powered to perform these subgroup analyses for different

amyloid types. Similarly, 'heart failure' is very broad term and is often miscoded. Further, it is HFPEF that we are interested in, and not just any heart failure.

4) Many of the phenotypes/predictors of wild-type amyloid that the authors find from the medical claims data are already well described in the literature and hence not new. The authors should discuss the specific benefits of using EHR data to make these associations, ie. many organ systems involved and overlap with many other medical conditions; hence ML can automatically identify these patterns that are not obvious to the clinician.

5) NRI and IDI was calculated and at least one should be included in the results

Reviewer #5 (Remarks to the Author):

300 million US patients from 1/2008 to 4/30/2019 from IQIVIA and 88 million from OPTUM using ICD codes. Matching based on HF and propensity score including age, gender, duration of medical history in DB, and number of healthcare visits. This is an interesting study with limitations that the authors readily admit.

1. The biggest concern is that that a sizable number of the controls may have undiagnosed amyloidosis. The authors mention it as a limitation in the "limitations" paragraph, but that is not sufficient. They should address it in the results/discussion and discuss how using models with higher specificity in this instance may be inferior to those with higher sensitivity.

2. In the model described page 10, how did they deal with the differential use of BNP and NT-proBNP and differential use of troponin I, troponin T, and hs-troponin T? Definition of high is different in Text and in Table 5 legend (0.4 in former and 0.04 in latter). The latter is correct for abnormal whereas the former is not.

Minor issues.

1. Supplement page 9, the decimal points are missing from the ICD codes.

2. Please define “recall” in Online Figure 2. Many MDs will not know that is same as sensitivity. What is F1 and cost-matrix gain? For online Table 4, also define Precision as PPV, recall as sensitivity for the MD readership.

3. In legend of Table 6, please define the two models. Paper is so dense, it is nice to have it accessible.

Reviewer #6 (Remarks to the Author):

Overall, I think the questions raised by reviewer #3 have been answered satisfactorily by the authors. To my mind the most important problem with this manuscript is that raised by reviewer #3's point below.

1. The obvious point – the model performance is predicated upon assignment of cases and controls. We can presume for the sake of argument that cases of ATTRwt-CM are likely to be accurate. But similar to the point made in the discussion regarding the NW dataset, there definitively unrecognized cases of ATTRwt-CM in the control cohort. This suggests that the performance would be actually better if the case and control cohorts were purely comprised, but since you are working from codes and non-rigorously adjudicated cases and controls, this is a weakness.

It is very likely that there will indeed be undiagnosed cases of ATTR-CM in the control cohort. Although the reviewer suggests that this is likely to mean the true performance would be better than that presented, I am not convinced that this is the case since the false negative rate will increase if this is the case. As suggested by the reviewer, this is an important weakness and ideally the negative controls should come from a cohort who have all undergone PYP scintigraphy (which all but excludes cardiac ATTR amyloidosis). Given that this undoubtedly compromises the assessment of accuracy of the algorithm, the editors may wish to consider requesting the addition of a more fully characterised control cohort.

Reviewer #7 (Remarks to the Author):

The responses to reviewer #4 are acceptable. I do strongly encourage to not perpetuate the use of AUROC as a metric in such situations and instead focus on AUPRC values, the increase in post-test PPV etc.

It does us all a disservice to play up AUROC, and numbers in the 80s, when they are practically useless in the use case envisioned and for low prevalence events can misguide a lot of readers.

The use case is about using a model as a screening tool (and not a diagnostic classifier). Therefore the exposition should begin and end with that. Yes, the ppv is low per se, but a 10-25x increase in finding a treatable condition which is often "missed" is real value added to medicine.

However that increased case finding ability needs to be balanced with the harm that will be inflicted on the false positives. Will they be tested with a "confirmatory test"? How much does that cost, is it invasive? And net-net what is the number needed to test (and \$\$ needed to spend) to find one new case of ATTR-CM, which we are currently missing.

That kind of discussion is essential.

RESPONSES TO REVIEWERS

Manuscript title: *A Machine Learning Model of Medical Claims Data for Identification of Wild-Type Transthyretin Amyloid Cardiomyopathy*

Manuscript #: NCOMMS-20-22672-T

Responses to Reviewer #2:

The study population, feature selection, and machine learning methodology used in this paper are very robust. The reviewers' comments have been somewhat addressed. It would be helpful to see changes made in the paper, this is currently missing. There are still major concerns which should be addressed throughout the manuscript carefully.

We thank the reviewer for their positive comments about our study. In the prior revision, we had uploaded a tracked changes version of the paper for the reviewers. We have made sure with this revision that all of the changes made in the paper also appear in the tracked changes version and have also outlined the changes in this response to reviewers document.

- 1. The clinical relevance of all this is unclear. The argument is that amyloid is often missed or not considered as an etiology of heart failure, and that there is no specific test for amyloid which is true. However wild-type amyloid is more often and indolent disease with a slower course, and it's not clear why this was specifically chosen to examine from the vast database.**

We believe that our study is clinically relevant, and the rationale for our study is clearly stated in the introduction of our paper. Wild-type ATTR cardiomyopathy is often underdiagnosed and missed clinically. If it is not diagnosed, it cannot be treated, and if it cannot be treated patients die of the disease. Thus, any algorithm or test that results in earlier and greater frequency of diagnosis would be expected to result in earlier treatment and improved outcomes. Now that wild-type ATTR cardiomyopathy has a proven treatment (tafamidis, a TTR stabilizer) which has been shown to reduce mortality and cardiovascular hospitalizations, we need methods to improve diagnosis. It is for these reasons that we sought to study wild-type ATTR cardiomyopathy.

Of note, the reviewer's statement that "wild-type amyloid is more often an indolent disease with a slower course" is not correct. When untreated survival of wild-type ATTR cardiomyopathy is very poor. For example, as shown in the figure below in a paper by Connors LH, et al (Amyloid 2011;18(Suppl 1):157-159, median survival was only 4.3 years, and worse than hereditary forms of ATTR cardiomyopathy other than patients with the *TTR* V122I variant (whose survival as not different than wild-type ATTR cardiomyopathy [$p=0.225$]). Note: in the figure below wild-type ATTR cardiomyopathy is abbreviated "SSA" (senile systemic amyloidosis).

PATIENTS, *n*

SSA	71	58	31	16	5	1	0	0	0
ATTR-V122I	30	25	11	4	1	0	0	0	0
ATTR-S77Y	11	11	10	6	5	3	2	1	0
ATTR-T60A	35	30	21	12	9	7	4	2	0

- Also, absence of monoclonal protein in serum combined with a positive bone scintigraphy can now diagnosis wild type ATTR with 100% specificity and PPV. The authors also fail to capitalize on their findings in their final discussion paragraph, as ultimately this machine learning model could work in the background to calculate prediction scores and flag patients with heart failure on EHR systems, and there is only brief mention of this.

We thank the reviewer for this comment and have added the following to our discussion section:

“We have developed and validated a machine learning approach based on medical claims data that provides a systematic framework to identify potential wild-type ATTR-CM patients (Figure 4), which could work in the background to automatically calculate probability scores for wild-type ATTR-CM and flag HF patients in EHR systems that may benefit from further evaluation for ATTR-CM.”

- ICD coding for wild-type amyloid was used as the gold standard – this is flawed as we do not know what the specific ECG, imaging, and histology findings are for these patients, or what criteria was used to make this diagnosis. Even within wild-type amyloid there are many different genetic variants all with different phenotypes, and the authors fail to tease this out as their ICD coding was purely “wild-type”. This study seems adequately powered to perform these subgroup analyses for different amyloid types. Similarly, ‘heart failure’ is very broad term and is often miscoded. Further, it is HFPEF that we are interested in, and not just any heart failure.

We had previously acknowledged in our paper that ICD coding is not the gold standard for the diagnosis and had discussed this extensively in our discussion section.

We agree that “heart failure” is a broad term and can be miscoded and have added this as a potential limitation in our discussion section as follows: “...our study is limited by the

use of ICD codes to identify HF patients and as the gold standard for assigning cases and controls. HF, a broadly defined clinical syndrome, may be inaccurately coded in the medical record.”

We disagree with the reviewer’s other statements:

- “Even within wild-type amyloid there are many different genetic variants all with different phenotypes, and the authors fail to tease this out as their ICD coding was purely “wild-type”.” This is incorrect. By definition wild-type ATTR cardiomyopathy is diagnosed in the *absence* of genetic variants in the *TTR* gene.
- “Further, it is HFpEF that we are interested in, and not just any heart failure.” This is also incorrect. It is a common misconception that wild-type ATTR cardiomyopathy causes only HFpEF and not heart failure with reduced ejection fraction. In fact, many patients with ATTR cardiomyopathy have a reduced ejection fraction. For example, in the ATTR-ACT randomized controlled trial which compared tafamidis vs. placebo in ATTR cardiomyopathy (Maurer M, et al. NEJM 2018), 22% of the patients in the trial had overt HFrEF (LV ejection fraction < 40%), 29% of patients had HF with mid-range ejection fraction (LV ejection fraction 40-50%), and 50% of patients had HFpEF (Pfizer data on file).

- 4. Many of the phenotypes/predictors of wild-type amyloid that the authors find from the medical claims data are already well described in the literature and hence not new. The authors should discuss the specific benefits of using EHR data to make these associations, ie. many organ systems involved and overlap with many other medical conditions; hence ML can automatically identify these patterns that are not obvious to the clinician.**

We thank the reviewer for this suggestion and have added the following sentence to our discussion section:

“Although many of the identified phenotypes/features associated with wild-type ATTR-CM that we identified in the medical claims data are already well described in the literature, a strength of our ML approach is the ability to automatically identify patterns suggestive of ATTR-CM which may not be obvious to the clinician, because they are derived using data from multiple diagnostic codes across several different disease domains and organ systems.”

- 5. NRI and IDI was calculated and at least one should be included in the results.**

We have now included NRI and IDI in the results section as follows:

“In addition, adding the ML algorithm-predicted probability of ATTR-CM to a base model that includes age, sex, race, and logBNP resulted in an integrated discrimination index of 0.039 (95% CI 0.031-0.048), $P < 0.00001$ and a category-less net reclassification index of 0.25 (95% CI 0.22-0.28), $P < 0.00001$, both indicative that the addition of the ML algorithm improved performance of a model that used conventional clinical predictors alone.”

Responses to Reviewer #5:

300 million US patients from 1/2008 to 4/30/2019 from IQIVIA and 88 million from OPTUM using ICD codes. Matching based on HF and propensity score including age, gender, duration of medical history in DB, and number of healthcare visits. This is an interesting study with limitations that the authors readily admit.

We thank the reviewer for these positive comments about our paper.

- 1. The biggest concern is that that a sizable number of the controls may have undiagnosed amyloidosis. The authors mention it as a limitation in the “limitations” paragraph, but that is not sufficient. They should address it in the results/discussion and discuss how using models with higher specificity in this instance may be inferior to those with higher sensitivity.**

We have added the following to our discussion section (in addition to what was already written in our limitations section):

“Nonetheless, it is important to note that future prospective studies, with systematic application of gold standard tests for the diagnosis of ATTR-CM will need to be curated for further development and validation of our ML model.”

- 2. In the model described page 10, how did they deal with the differential use of BNP and NT-proBNP and differential use of troponin I, troponin T, and hs-troponin T? Definition of high is different in Text and in Table 5 legend (0.4 in former and 0.04 in latter). The latter is correct for abnormal whereas the former is not.**

In the single-center Northwestern (NMEDW) validation cohort the BNP and troponin assays are uniform (we only use a single assay for both tests at Northwestern Memorial Hospital). These are BNP (not NTproBNP), and troponin I (not troponin T or hs-troponin T). We have corrected the error in the troponin thresholds (it should have been >0.04 ng/ml in both the text and the table).

- 3. Minor issues:**

- Supplement page 9, the decimal points are missing from the ICD codes.**

The decimal points have been added to the ICD codes.

- Please define “recall” in Online Figure 2. Many MDs will not know that is same as sensitivity. What is F1 and cost-matrix gain? For online Table 4, also define Precision as PPV, recall as sensitivity for the MD readership.**

We have made these changes in Online Figure 2 (which is now Supplementary Figure 1 in the revised Supplementary Materials section). We have also made these changes to Online Table 4 (which is now Supplementary Table 5 in the revised Supplementary Materials section).

- In legend of Table 6, please define the two models. Paper is so dense, it is nice to have it accessible.**

We have added the definition of the 2 models in the legend of Table 6 as follows:

“The random forest ATTR-CM model was derived using diagnosis codes specifically for wild-type ATTR-CM. The random forest cardiac amyloid model was derived using the more non-specific umbrella diagnosis code for cardiac amyloidosis.”

Responses to Reviewer #6:

Overall, I think the questions raised by reviewer #3 have been answered satisfactorily by the authors. To my mind the most important problem with this manuscript is that raised by reviewer #3's point, “The obvious point – the model performance is predicated upon assignment of cases and controls. We can presume for the sake of argument that cases of ATTRwt-CM are likely to be accurate. But similar to the point made in the discussion regarding the NW dataset, there definitively unrecognized cases of ATTRwt-CM in the control cohort. This suggests that the performance would be actually better if the case and control cohorts were purely comprised, but since you are working from codes and non-rigorously adjudicated cases and controls, this is a weakness.”

- 1. It is very likely that there will indeed be undiagnosed cases of ATTR-CM in the control cohort. Although the reviewer suggests that this is likely to mean the true performance would be better than that presented, I am not convinced that this is the case since the false negative rate will increase if this is the case. As suggested by the reviewer, this is an important weakness and ideally the negative controls should come from a cohort who have all undergone PYP scintigraphy (which all but excludes cardiac ATTR amyloidosis). Given that this undoubtedly compromises the assessment of accuracy of the algorithm, the editors may wish to consider requesting the addition of a more fully characterised control cohort.**

We acknowledge the limitations of our dataset in identifying cases and controls based on ICD codes. Unfortunately, we are not aware of any large-scale cohorts of heart failure patients who have been consecutively and systematically examined with confirmatory tests such as PYP scintigraphy to rule in or rule out wild-type ATTR cardiomyopathy. We see our algorithm as a starting point that could stimulate such studies in the future.

Responses to Reviewer #7:

- 1. The responses to reviewer #4 are acceptable. I do strongly encourage to not perpetuate the use of AUROC as a metric in such situations and instead focus on AUPRC values, the increase in post-test PPV etc. It does us all a disservice to play up AUROC, and numbers in the 80s, when they are practically useless in the use case envisioned and for low prevalence events can misguide a lot of readers.**

We have removed the listing of AUROC values from the abstract. We did not include AUPRC values because we feel that they are difficult to interpret for clinicians (due to the lack of a standard lower limit threshold such as 0.5 for AUROC), and the post-test probability likelihood ratios (as listed in Table 5 in the revised manuscript) should be useful to clinicians to help understand the clinical utility of our algorithm.

We do agree that reporting AUROC has its limitations. We now list the limitations of using the AUROC in our discussion section as follows:

“...AUROC may be misleading for relatively rare (i.e., low prevalence) diagnoses such as wild-type ATTR-CM. However, we report a wide range of model performance indices in addition to the AUROC, and we include positive and negative likelihood ratios (Table 5), which are useful for the clinical interpretation of our results.”

- 2. The use case is about using a model as a screening tool (and not a diagnostic classifier). Therefore the exposition should begin and end with that. Yes, the ppv is low per se, but a 10-25x increase in finding a treatable condition which is often "missed" is real value added to medicine.**

We thank the reviewer for this suggestion. We have now indicated that our goal was to create a screening tool (and not a diagnostic tool). We have made the following edits to the introduction and discussion section of our paper:

Introduction: “Given the underdiagnosis and significant morbidity of ATTR-CM and availability of treatment with TTR stabilization, improved screening to enhance early diagnosis of the disease is essential...” and “We sought to develop and validate a machine learning model based on administrative medical claims data in the electronic health record with the goal of creating a resource to facilitate the systematic screening and identification of patients with wild-type ATTR-CM

Discussion: “We have developed and validated a machine learning approach based on medical claims data that provides a systematic framework for the screening and identification of potential wild-type ATTR-CM patients (Figure 4),...”

- 3. However that increased case finding ability needs to be balanced with the harm that will be inflicted on the false positives. Will they be tested with a "confirmatory test"? How much does that cost, is it invasive? And net-net what is the number needed to test (and \$\$ needed to spend) to find one new case of ATTR-CM, which we are currently missing. That kind of discussion is essential.**

The reviewer raises an important point. The increase case finding ability of this model may indeed lead to confirmatory testing, which would include evaluation for light-chain amyloidosis (blood and urine tests) coupled with either noninvasive ^{99m}Tc-PYP cardiac scintigraphy or in approximately ~20% of cases, invasive endomyocardial biopsy, and in those with positive results, TTR genetic testing. These costs must be balanced with the quality of life years saved due to early diagnosis and treatment of ATTR-CM patients, along with the potential for reduced costs if patients with ATTR-CM are diagnosed and treated earlier in the course of disease, which could result in lower costs for hospitalizations and emergency room visits. A formal cost effectiveness evaluation of our machine learning-based diagnostic algorithm is outside of the scope of our current analysis and premature; further prospective validation of our machine learning algorithm will be necessary, and if validated, overall cost implications and analysis should be pursued.

Considering the above potential risks of testing, the risk benefit profile favors potential finding of a lethal cardiac condition that is now treatable. It is important to note, however,

that the case finding ability of this algorithm ultimately flags individuals potentially at-risk for wild-type ATTR-CM but does not diagnose the condition. Ultimately, confirmatory diagnostic testing remains at the discretion of the clinical judgment of the treating physician, who may choose to follow-up on this flag through more thorough past medical history and/or chart review in order to modify pretest probability.

We have now included the following sentence in our discussion section:

“Nevertheless, the potential for improved early diagnosis of wild-type ATTR-CM with incorporation of these ML algorithms must be balanced with the potential cost implications and downstream effects of false positives.”

REVIEWERS' COMMENTS

Reviewer #7 (Remarks to the Author):

My comments are adequately addressed.

Overall, this paper gets to the heart of the common tension between classical statistical analyses and modern machine learning. The tension was identified in ~2000 by Leo Breiman in his classic paper on 'Two Cultures' and recently again by Brad Efron in <https://www.tandfonline.com/doi/abs/10.1080/01621459.2020.1762617>.

Reviewer #8 (Remarks to the Author):

No further comments

Reviewer #9 (Remarks to the Author):

Authors have made a valiant effort to address the limitations cited by the previous reviewers. There is indeed value to this paper in potentially screening the population for Wild-type TTR Amyloidosis, however, I do agree with previous reviewers' suggestions that 'the algorithm only flags individuals potentially at-risk for wild-type ATTR-CM but does not diagnose the condition'. This is not forthcoming and the manuscript title and results may mislead a reader.

1. I would recommend the title be modified as: "A Machine Learning Model of Medical Claims Data for Identifying Patient Population at Risk for Wild-Type Transthyretin Amyloid Cardiomyopathy"
2. In the conclusions, I would specifically mention, "Despite the constraints of the model being built on a medical claim database, we illustrated that the ML model may be potentially useful in identifying ATTRwt-CM..."
3. I would tone down all throughout the manuscript carefully any claims towards diagnosis and early detection of ATTRwt-CM"

RESPONSES TO REVIEWERS

Manuscript title: *A Machine Learning Model of Medical Claims Data for Identifying Patients at Risk for Wild-Type Transthyretin Amyloid Cardiomyopathy*
Manuscript #: NCOMMS-20-22672A

Responses to Reviewer #7:

1. **My comments are adequately addressed. Overall, this paper gets to the heart of the common tension between classical statistical analyses and modern machine learning. The tension was identified in ~2000 by Leo Breiman in his classic paper on 'Two Cultures' and recently again by Brad Efron in <https://www.tandfonline.com/doi/abs/10.1080/01621459.2020.1762617>.**

We thank the reviewer for their comments and support of our manuscript.

Responses to Reviewer #8:

1. **No further comments**

We thank the reviewer for their comments and support of our manuscript.

Responses to Reviewer #9:

1. **Authors have made a valiant effort to address the limitations cited by the previous reviewers. There is indeed value to this paper in potentially screening the population for Wild-type TTR Amyloidosis, however, I do agree with previous reviewers' suggestions that 'the algorithm only flags individuals potentially at-risk for wild-type ATTR-CM but does not diagnose the condition'. This is not forthcoming and the manuscript title and results may mislead a reader.**

We thank the reviewer for their comments and support of our manuscript. We agree with the reviewer that we could do a better job stating clearly that our algorithm only flags individuals potentially at-risk for wild-type ATTR-CM and have addressed this concern in our response to the reviewer, as detailed below.

2. **I would recommend the title be modified as:"A Machine Learning Model of Medical Claims Data for Identifying Patient Population at Risk for Wild-Type Transthyretin Amyloid Cardiomyopathy"**

We have revised our title as follows: "A Machine Learning Model of Medical Claims Data for Identifying Patients at Risk for Wild-Type Transthyretin Amyloid Cardiomyopathy"

3. **In the conclusions, I would specifically mention, "Despite the constraints of the model being built on a medical claim database, we illustrated that the ML model may be potentially useful in identifying ATTRwt-CM..."**

We have edited the 2nd sentence in our conclusions paragraph to state the following:

“Despite the constraints of using medical claims data to build our prediction model, we illustrated that the machine learning model may be potentially useful in identifying wild-type ATTR-CM patients and could work in the background in EHR systems to automatically calculate probability scores and flag HF patients that may benefit from further evaluation for ATTR-CM.”

4. **I would tone down all throughout the manuscript carefully any claims towards diagnosis and early detection of ATTRwt-CM**

In prior revisions of our manuscript, we had toned down any claims towards diagnosis and early detection of ATTRwt-CM, and our manuscript already states the following:

“Finally, our model should not be viewed as definitive for the diagnosis of either ATTR-CM or cardiac amyloidosis, but instead could be used as a starting point for identifying at-risk individuals for further evaluation with speckle-tracking echocardiography, cardiac MRI with contrast, bone scintigraphy, and/or blood tests for amyloidogenic light chain amyloidosis. Subsequent evolution and simplification of this ML algorithm might further be clinically applicable at the individual patient level, for instance, in a HF clinic setting, to help assess probability and likelihood of ATTR-CM among patients presenting with undifferentiated HFpEF.”

We carefully reviewed our manuscript and found 2 additional instances where we could further tone down claims towards detection and early diagnosis.

We have changed the following text in the discussion section: “The results of our study are important because our predictive model, if successfully employed within the EHR of healthcare systems, could lead to **earlier identification, diagnosis,** and treatment of patients with wild-type ATTR-CM (and potentially other forms of cardiac amyloidosis) by raising the suspicion of the diagnosis in HF patients.”

The revised text reads as follows:

“The results of our study are important because our predictive model, if successfully employed within the EHR of healthcare systems, could lead to targeted testing and confirmation of ATTR-CM in at-risk patients, which could lead to earlier treatment of patients with wild-type ATTR-CM (and potentially other forms of cardiac amyloidosis) by raising the suspicion of the diagnosis in HF patients.”

In addition, we have added the word “ultimately” prior to “diagnosed” in the following sentence in our discussion section:

“Furthermore, even if a high probability threshold for the diagnosis of ATTR-CM was used (e.g., > 0.75), as shown in **Table 3**, nearly 10% additional patients with ATTR-CM could be ultimately diagnosed, which would be an important clinical advance.”